# *Bauhinia* (Leguminosae) Fossils from the Paleogene of Southwestern China and Its Species Accumulation in Asia

**Lin-Bo Jia** [1,*,†][iD], **Jin-Jin Hu** [1,†][iD], **Shi-Tao Zhang** [2], **Tao Su** [3], **Robert A. Spicer** [3,4], **Jia Liu** [3], **Jiu-Cheng Yang** [2][iD], **Pu Zou** [5,*], **Yong-Jiang Huang** [1] **and Zhe-Kun Zhou** [3]

1   CAS Key Laboratory for Plant Diversity and Biogeography of East Asia, Kunming Institute of Botany, Chinese Academy of Sciences, Kunming 650201, China; hujinjin@mail.kib.ac.cn (J.-J.H.); huangyongjiang@mail.kib.ac.cn (Y.-J.H.)
2   Faculty of Land Resource Engineering, Kunming University of Science and Technology, Kunming 650093, China; taogezhang@hotmail.com (S.-T.Z.); 18395636983@163.com (J.-C.Y.)
3   CAS Key Laboratory of Tropical Forest Ecology, Xishuangbanna Tropical Botanical Garden, Chinese Academy of Sciences, Mengla 666303, China; sutao@xtbg.org.cn (T.S.); ras6@open.ac.uk (R.A.S.); liujia@xtbg.ac.cn (J.L.); zhouzk@xtbg.ac.cn (Z.-K.Z.)
4   School of Environment, Earth and Ecosystem Sciences, The Open University, Milton Keynes MK7 6AA, UK
5   South China Botanical Garden, Chinese Academy of Sciences, Guangzhou 510650, China
*   Correspondence: jialinbo@mail.kib.ac.cn (L.-B.J.); zoupu@scbg.ac.cn (P.Z.)
†   These authors contributed equally to this work.

**Abstract:** Extant *Bauhinia* (Leguminosae) is a genus of 300 species of trees, shrubs, and lianas, widely distributed in pantropical areas, but its diversification history in southeastern Asia, one of its centers of highest diversity, remains unclear. We report new fossils of three *Bauhinia* species with cuticular preservation from the Paleogene of Puyang Basin, southwestern China. Our finding likely extends the emergence of *Bauhinia* in Asia to the late Eocene. Together with previously reported fossil records, we show that the diversification of *Bauhina* in Asia and the phenomenon of a small region harboring multiple *Bauhinia* species in southwestern China could be traced back to the Paleogene.

**Keywords:** Asia; Biogeography; *Bauhinia*; Fabaceae; late Eocene

## 1. Introduction

*Bauhinia* L. (Leguminosae) today comprises about 300 species of trees, shrubs, and lianas and is widely distributed in pantropical areas, with the largest diversity center being in the neotropics, and the second largest in southeastern Asia [1–3] (Figure 1). A typical leaf of this genus is simple and bilobed, rarely entire or two-foliolate, with pulvinus on both ends of the petiole [1]. Its fruit is flat, elliptic, oblong, or linear, woody or thinly valved. Species of *Bauhinia* are widely cultivated as ornamentals [1]. For example, the orchid tree (*Bauhinia* × *blakeana* Dunn) was chosen as the city flower of Hongkong. Several species of *Bauhinia* (e.g., *B. purpurea* L.) are used in local medicine and seeds of *B. petersiana* Bolle can be used as a coffee substitute [3].

Recent phylogenetic studies show that *Bauhinia* is an early-diverged member of Leguminosae [2,4–7]. However, due to the nesting of *Griffonia* and *Brenierea* within the genus, *Bauhinia* is not monophyletic [2,5]. The phylogenetic relationships of the *Bauhina* + *Griffonia* + *Brenierea* clade has not yet been well resolved [2,5]. In some recent treatments, *Bauhinia* is divided into eight genera including *Bauhinia* L. s.s., *Barklya* F. Muell., *Gigasiphon* Drake, *Lysiphyllum* (Benth.) de Wit, *Phanera* Lour., *Piliostigma* Hochst., *Schnella* Raddi, and *Tylosema* (Schweinf.) Torre et Hillc. [5,8]. Here we adopt the traditional broad treatment of *Bauhinia* because this study mainly concerns plant morphology and the character suite available in *Bauhinia* leaf fossils limits taxonomic resolution.

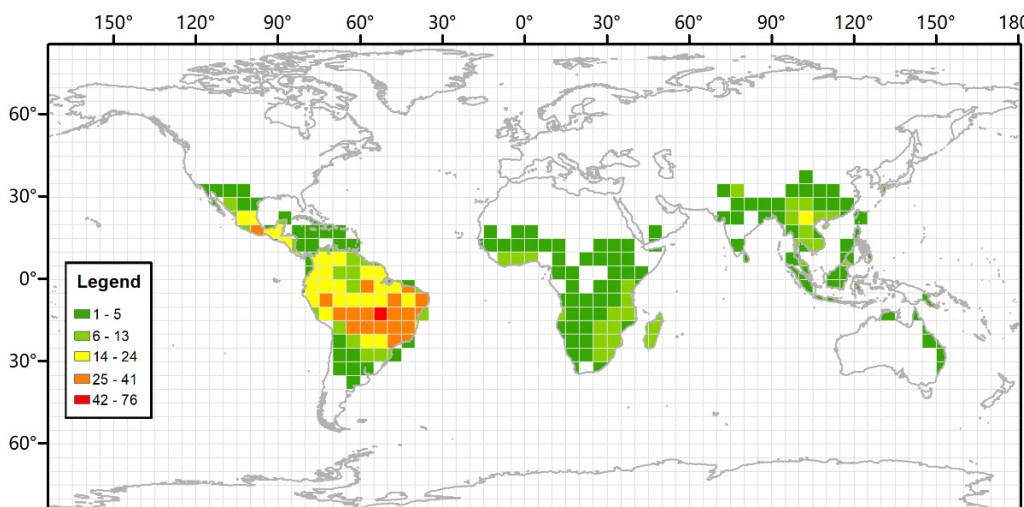

**Figure 1.** Extant distribution of *Bauhinia*. Different colors in the map indicate the number of species in each grid square. Extant occurrence data of *Bauhinia* are from the Global Biodiversity Information Facility (GBIF).

Fossils of *Bauhinia* have been documented in various forms of wood, leaf, and twig with attached fruit [9–12]. The earliest reliable fossils are leaves from the early Oligocene of China [11,13]. Later fossils of the genus are documented from the late Oligocene and Middle Miocene of China, the Oligocene of Mexico, and the early Miocene and middle Miocene–middle Pleistocene of India, the Pliocene of Uganda, and the Miocene of Ecuador [9,10,12,14–18].

This study reports new *Bauhinia* fossils from the late Eocene of southeastern China. First, we morphologically compared the macroscopic morphology and cuticle features of the fossils with those of the extant and fossil species in the genus. Then we discussed the implications of the fossils in the context of our current understanding of the evolutionary history of *Bauhinia* in Asia.

## 2. Materials and Methods

### 2.1. Geological Setting

The Puyang Basin (105.26° E, 23.48° N; 825 m asl) is a wedge-shaped strike-slip basin located in the southeastern Yunnan province, China [19–21] (Figure 2). The base of the basin is Cambrian limestone, with Cenozoic sediments unconformably lain above [20] (Figure 3). The lower part of the Cenozoic basin fill is dominated by lignite beds representing swamp facies, and the upper part is mainly lacustrine grey to yellow mudstone [20]. Recently, a mammal fossil, belonging to Anthracotheriidae Leidy, similar to the late Eocene *Bothriogenys hui* from Yunnan and *B. orientalis* from Thailand [22,23], was recovered from the lignite (Figure 3). Pollen analysis also suggests a late Eocene age for the lignite bed (Yang et al. under review) and suggests that basin formation was roughly coeval with other regional basins such as Wenshan [13] and Lühe [24] which have been dated radiometrically. Our fossils are collected from the lacustrine mudstone above the lignite bed (Figure 3) and are most likely also late Eocene in age.

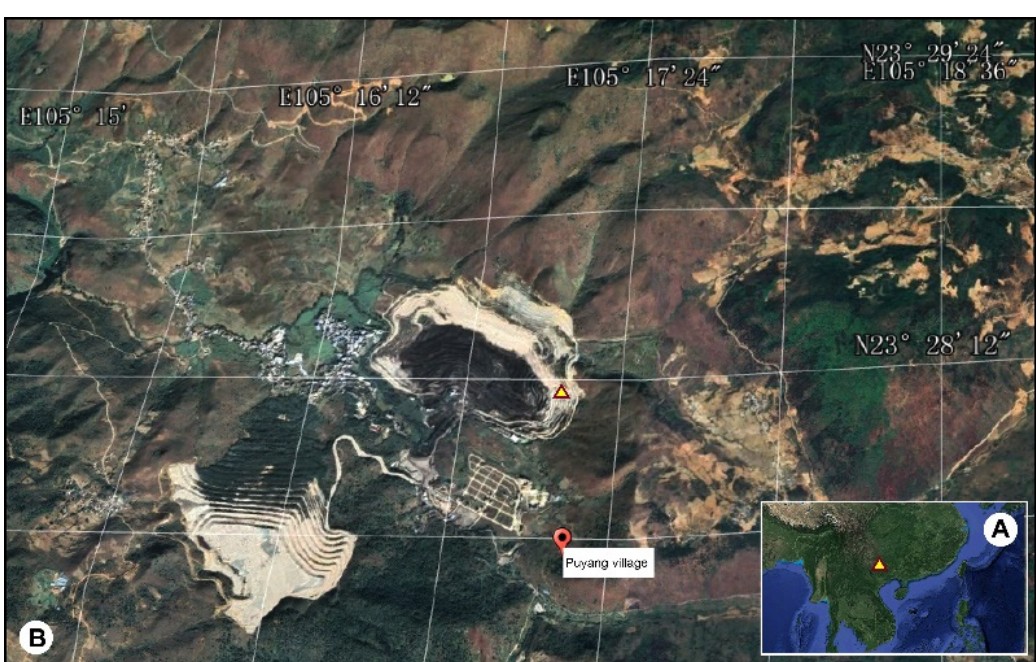

**Figure 2.** The position of the fossil locality in a broad view of southeastern Asia (**A**) and in a magnified view of Puyang Basin, Yunnan province, China (**B**).

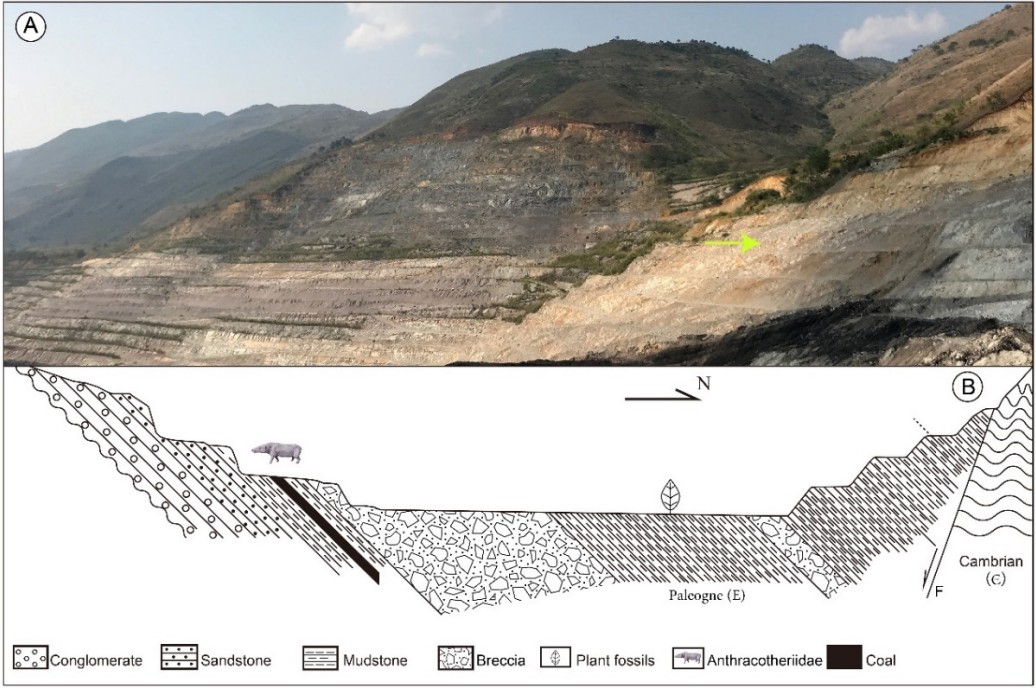

**Figure 3.** Cross section of the fossil locality in Puyang Basin, southeastern Yunnan Province, China. (**A**) The fossiliferous outcrop. The light green arrow indicates the layers where our fossils were collected. (**B**) Cross section of the fossil locality.

### 2.2. Macroscopic Feature Observations and the Modern Distribution of Bauhinia

Fifteen leaf fossils and one fruit compression were recovered and photographed using a digital camera (Nikon D750, Kanagawa, Japan). Fine-scale details of the fossils were further examined under a stereo microscope (Leica S8APO, Wetzlar, Germany), and images were taken. The raw data for the extant occurrence of *Bauhinia* was download from Gbif [25],

and first cleaned using an R program and then checked manually [26]. Finally, the cleaned data were imported into Arcgis 10.0 to prepare the distributional heat map.

### 2.3. Cuticle Preparation for Fossil and Extant Materials

Fossil leaf fragments were treated with HCl and HF to remove calcareous and siliceous materials, and then macerated using 3% NaClO solution for 30 min to one hour until they became translucent [27–29]. For extant materials, fragments from mature leaves were macerated using a 1:1 solution of $CH_3COOH$ and 30% $H_2O_2$ at 80 °C for about one hour [30,31]. After the mesophyll tissue was removed, the adaxial and abaxial cuticles for both fossil and extant materials were stained for about 30 min using Safranin O, mounted in glycerine on glass slides, and then photographed using a light microscope (Leica DM 750 with a Leica DFC 295 camera). All cuticular slides are stored at Kunming Institute of Botany, Chinese Academy of Sciences.

### 3. Results

**Family**: Leguminosae Juss. (or Fabaceae Lindl).
**Subfamily**: Caesalpinioideae DC.
**Genus**: *Bauhinia* L.
**Locality**: The Puyang Basin, Funing county, Yunnan province, China.
**Age**: The late Eocene.
Leaf.
**Species**: *Bauhinia wenshanensis* H.H. Meng et Z.K. Zhou (morphotype 1).
2014 *Bauhinia wenshanensis* H.H. Meng et Z.K. Zhou, Figure 4A–D.
**Specimens**: FN0403 (Figure 4A); FN0399 (Figure 4B); FN0106 (Figure 4C); FN06005 (Figure 4D).

**Description**: Leaf is entire and bilobed, 28–34 mm long and 18–26 mm wide, ovate to elliptical in outline (Figure 4A–D). The basal portion is cordate, slightly asymmetrical (Figure 4A,B). The widest part is in the lower third of the leaf, and the lamina gradually tapers toward the apex (Figure 4A,B). The apex is bifid to form two acute lobes at an angle of 31°–49° (Figure 4A,B). The primary vein framework is palmate with nine basal veins (Figure 4A,B). Primary veins near the midvein extend into the apices of lobes (Figure 4A–C). Additional primary veins extend toward the adjacent primary vein at the inner side (Figure 4A,B). Major secondaries originate from the primary veins and extend toward the apex of the leaf (Figure 4A,B).

The adaxial cuticle consists of irregular epidermal cells with sinuolate epidermal walls (Figure 4E,F) and a few single-celled trichome bases (Figure 4E,G,H). No stomata were observed in the adaxial cuticle. The abaxial epidermal cells are polygonal or irregularly shaped. Stomatal complexes are paracytic and tetracytic with sunken guard cells, and the stomatal rim is single-layered (Figure 4I–K). Subsidiary cells are crescent, polygonal, or irregularly shaped. Many single-celled trichome bases exist in the abaxial cuticle (Figure 4I,L).

**Species**: *Bauhinia* sp. (morphotype 2).
**Specimens**: FN0411a (Figure 5A); FN0411b (Figure 5B); FN0292 (Figure 5C).

**Description**: Leaf is simple, petiolate and bilobed, 25–27 mm long and 17–31 mm wide, elliptical to obovate in outline (Figure 5A–C). Base is almost straight in open leaf (Figure 5A–C). Apex is round (Figure 5A,B). The primary vein framework is palmate with seven basal veins (Figure 5A–C).

The adaxial cuticle consists of polygonal epidermal cells with straight arched epidermal walls (Figure 5D–F) and few single-celled trichome bases (Figure 5G). No stomata were observed in the adaxial cuticle. The abaxial epidermal cells are similar in shape and size to those in the adaxial cuticle. Stomatal complexes are paracytic and tetracytic with sunken guard cells, and the stomatal rim is double-layered (Figure 5H–J). Subsidiary cells are polygonal or irregularly shaped. Many single-celled trichome bases exist in the abaxial cuticle (Figure 5H–K).

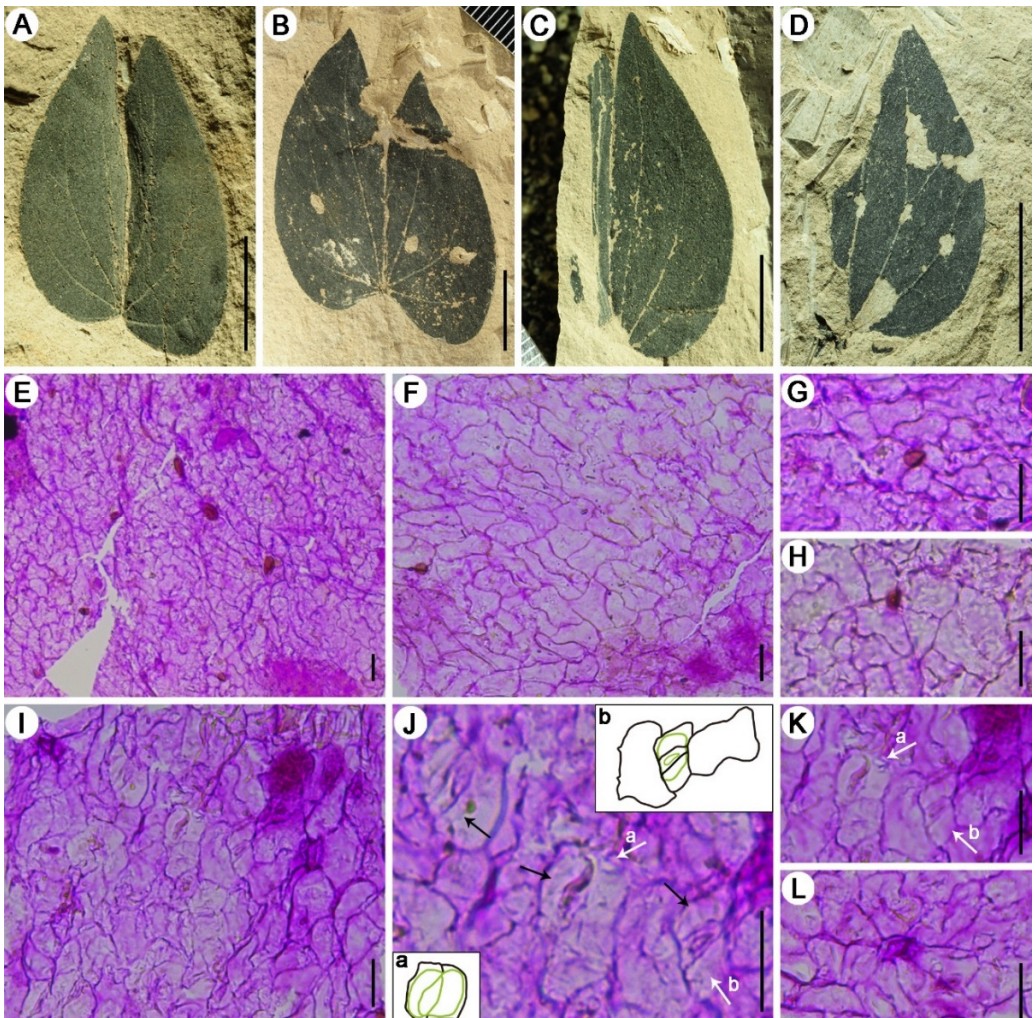

**Figure 4.** Leaf morphologies and cuticular structures of *Bauhinia wenshanensis* H.H. Meng et Z.K. Zhou (morphotype 1) from the Puyang Basin, Funing, Yunnan province. The scale bars represent 1 cm for leaf fossils and 20 μm for cuticle images. (**A**–**D**) Leaves of *B. wenshanensis*. All the cuticle images are from the fossil specimen in (**D**). (**E**,**F**) Adaxial cuticle showing sinuolate epidermal walls. (**G**,**H**) Single-celled trichome bases of adaxial cuticle. (**I**–**K**) Abaxial cuticle showing the orientation of stomata. Black arrows indicate sunken guard cells; white arrows indicate (a) paracytic stomatal complex, (b) tetracytic stomatal complex, and that the stomatal rim is single-layered. Line drawings illustrate (a) the paracytic stomatal complex indicated with white arrow a, (b) the tetracytic stomatal complex indicated with white arrow b. The green lines indicate stomata, and the black lines indicate subsidiary cells. (**L**) Single-celled trichome base of abaxial cuticle. (**A**), FN0403; (**B**), FN0399; (**C**), FN0106; (**D**), FN06005.

 **Species**: *Bauhinia* sp. (morphotype 3).
 **Specimens**: FN0189 (Figure 6A); FN0616 (Figure 6B).
 **Description**: Leaf is entire and bilobed, 18 mm long and 22 mm wide, elliptical to oblong in outline (Figure 6A,B). The basal portion is cordate and weakly asymmetrical (Figure 6B). The widest part is in the middle of the leaf (Figure 6A,B). Primary vein framework is palmate with seven basal veins (Figure 6B).
 Fruit.
 **Species**: cf. *Bauhinia* sp. (morphotype 4).
 **Specimens**: FN0465 (Figure 6C–E).
 **Description**: Fruit is flat, elliptic to oblong, 42 mm long and 17 mm wide (Figure 6C). The left flank of the proximal end is nearly straight, and the right flank is convex (Figure 6D).

The distal end is acuminate (Figure 6E). The stigmatic remain is short and persistent (Figure 6E). There is a constriction in the middle of the fruit (Figure 6C). The suture lines are prominent, about 0.5 mm wide (Figure 6D,E). The seed chambers are elliptic, 4.7–10.4 mm long and 3.6–4.6 mm wide (Figure 6C,E). The angle between the long axis of the seed chambers and those of fruit is 94–100° (Figure 6C).

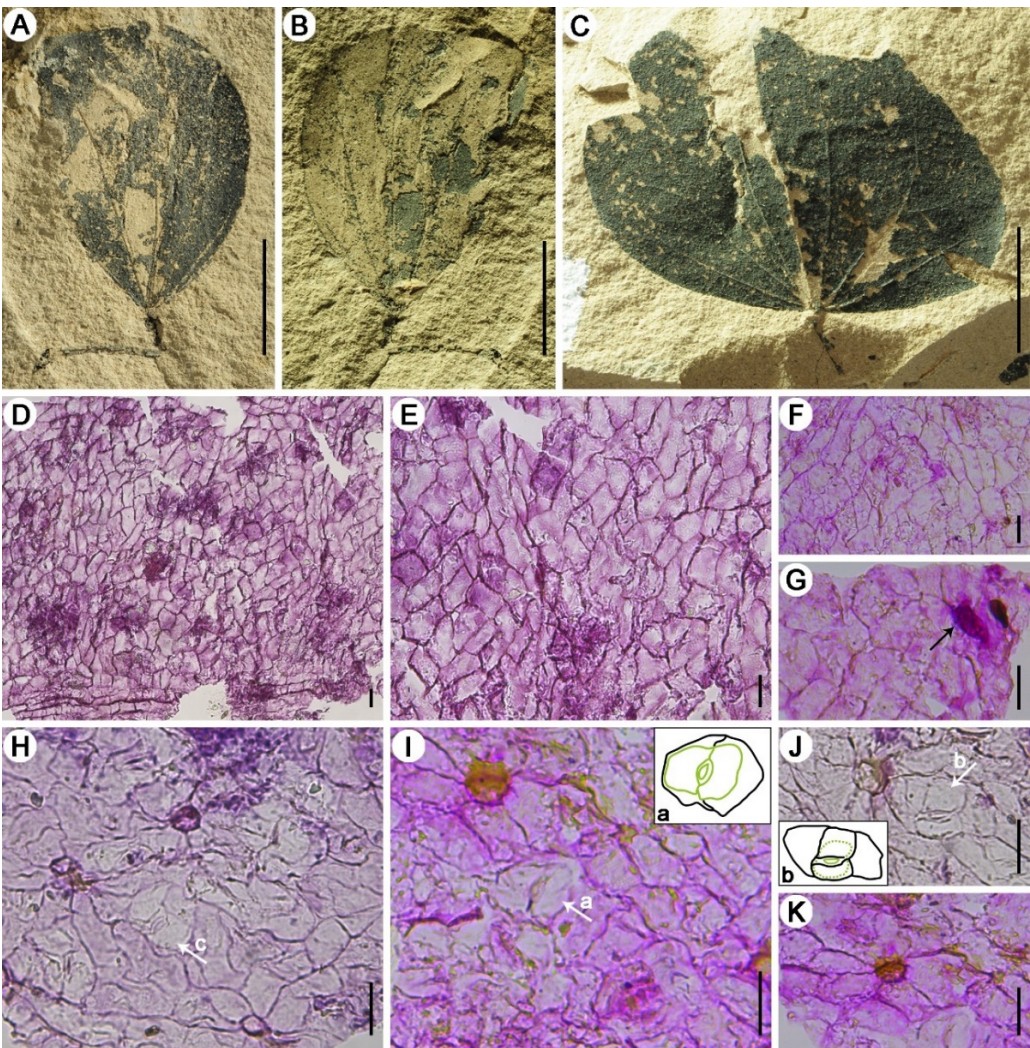

**Figure 5.** Leaf morphologies and cuticular structures of *Bauhinia* sp. (morphotype 2) from the Puyang Basin, Funing, Yunnan province. The scale bars represent 1 cm for leaf fossils and 20 μm for cuticle images. (**A–C**) Leaves of *Bauhinia* sp. The cuticle images in (**D,E,H,J**) are from the fossil specimen in (**C**); (**F,G,I,K**) are from the fossil specimen in (**A**). (**D–G**) Adaxial cuticle showing straight arched epidermal walls. The black arrow in (**G**) indicates a single-celled trichome base. (**H–J**) Abaxial cuticle showing the orientation of stomata. White arrows indicate (a) paracytic stomatal complex with double-layered stomatal rim, (b) tetracytic stomatal complex, and (c) stomatal complex with double-layered stomatal rim. Line drawings illustrate (a) the paracytic stomatal complex indicated with white arrow a, (b) the tetracytic stomatal complex indicated with white arrows b. The green lines indicate stomata, the dashed green lines indicate indistinct sunken guard cells, and the black lines indicate subsidiary cells. (**K**) Single-celled trichome base of abaxial cuticle. (**A**), FN0411a; (**B**), FN0411b; (**C**), FN0292.

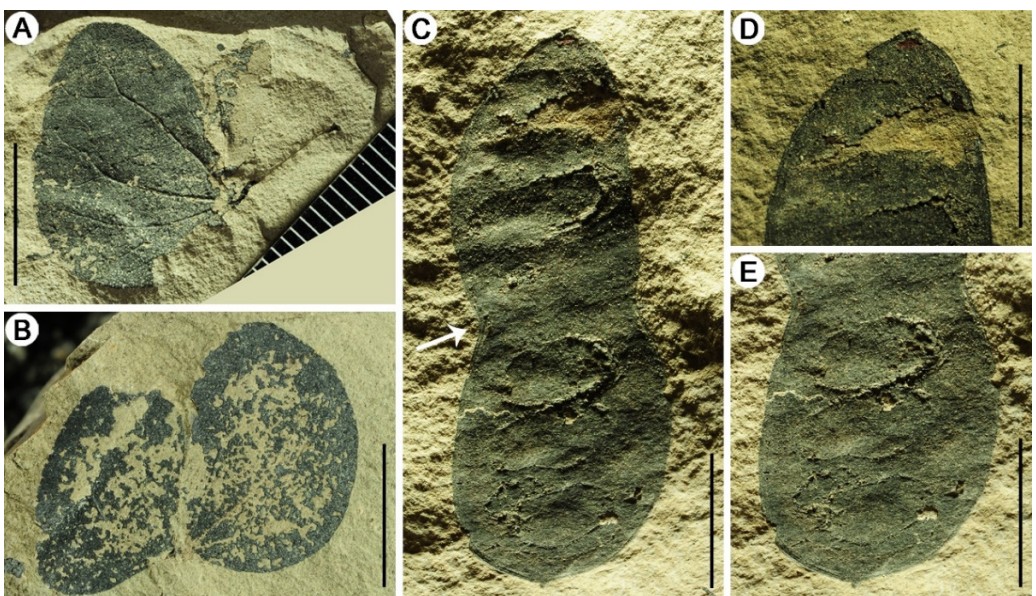

**Figure 6.** Leaf morphologies of *Bauhinia* sp. (morphotype 3) and fruit morphology of cf. *Bauhinia* sp. (morphotype 4). The scale bars represent 1 cm. (**A**) FN0189; (**B**) FN0616; (**C–E**) FN0465.

## 4. Discussion

### 4.1. Morphological Comparison

The fossil leaves are characterized by simple and bilobed leaves. As far as we know, such kinds of leaves are seen in several families including Ginkgoaceae Engl., Lauraceae Juss., Passifloraceae Juss. ex Roussel, Proteaceae Juss., and Leguminosae Juss. (Figure 7). However, venation of the Ginkgoaceae leaves is dichotomous, and so different from our fossils where the venation is reticulate. The leaves of Passifloraceae have a small middle lobe or a broad angle (larger than 90°) between the lobes, distinguishing them from our fossils, which are strictly bilobed and diverge at an angle less than 90°. The palmate venation of *Dilobia* Thours. in the Proteaceae (see cleared leaf in Pole and Bowman (1996) [32]) and some species in Lauraceae such as *Sassafras albidum* (Nutt.) Nees is suprabasal but that of our fossils is basal. In Leguminosae, *Pueraria* DC., *Desmodium* Desv., *Christia* Moench, and *Bauhinia* L. have bilobed leaves, but leaves of *Pueraria* are prominently deflective and asymmetric, differing from our fossils, which are nearly symmetric. Secondary veins of *Desmodium* leaves are parallel whereas those of our fossils extend towards the leaf apices. Leaves of *Christia* have pinnate venation, distinguishing them from our fossils which possess palmate venation. Overall, our fossil leaves are a close match with *Bauhinia*.

Based on macroscopic and cuticular morphology, the fossil leaves can be divided into three morphotypes. Morphotype 1 has a cordate base, acute apex, single-layered stomatal rim, and sinuolate adaxial epidermal walls. Morphotype 2 has a straight base, round apex, double-layered stomatal rim, and straight arched epidermal walls, whereas morphotype 3 has a cordate base and a round apex. Although leaf and epidermal cell shape display intraspecific variability, the characteristics of the stomatal rim are considered stable at intraspecific level [33,34], so morphotypes 1 and 2 should represent different species. Whether morphotype 3 is another species will be discussed below. The three morphotypes are further compared with 46 extant species based on macroscopic morphology and cuticle features (see images and tables in Zou [35]), and then they are compared with fossil species.

Morphotype 1 is similar to *B. acuminata* L., *B. comosa* Craib, and *B. esquirolii*; Gagnep. in gross macroscopic morphology (Figures 4 and 8). However, the abaxial cuticle of morphotype 1 has paracytic and tetracytic stomatal complexes with sunken guard cells (Figure 4J,K), and so is different from those of *P. comosa* and *P. esquirolii* that have paracytic stomatal complexes and the guard cells are not sunken (Figure 8F,I). The abaxial cuticle of morphotype 1 is similar to that of *B. acuminata* in features including paracytic and tetracytic

(some atypical) stomatal complexes with sunken guard cells and single-layered stomatal rim (Figure 8C). A combination of macroscopic and cuticular features suggests morphotype 1 is possibly a close relative of *B. acuminata*. However, it is worth noting that stomata also appear on the adaxial epidermis of *B. acuminata* in small number (Figure 8B), but we did not observe any from the adaxial epidermis of morphotype 1 (Figure 4E,F). This may be because the region from which we successfully extracted epidermis lacks stomata while the rest of leaf has them, or that morphotype 1 is a hypostomatic leaf. When compared with the reported fossil species (Table 1), morphotype 1 is similar to *B. wenshanensis* H. H. Meng et Z. K. Zhou found from the early Oligocene of Yunan, China [11]. In consideration of its near identical age, we assign morphotype 1 to *B. wenshanensis.* However, cuticular features have not yet been reported for *B. wenshanensis*, so our description of the cuticle for morphotype 1 may be taken as a tentative description for *B. wenshanensis*, but this should be used with caution. Future studies may obtain cuticle for *B. wenshanensis* from its type locality.

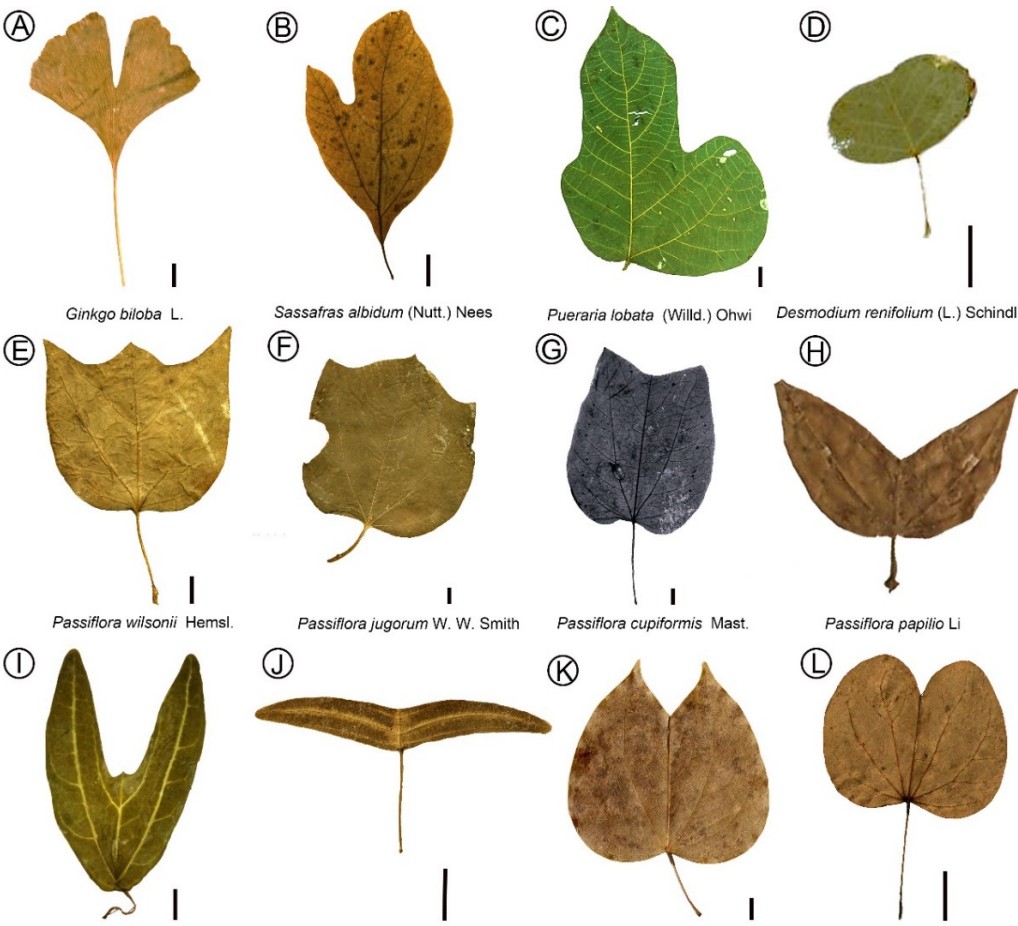

**Figure 7.** Leaf morphologies of bilobed or bilobe-like leaves in Ginkgoaceae Engl. (**A**) Lauraceae Juss. (**B**), Passifloraceae Juss. ex Roussel (**E**–**I**), and Leguminosae Juss. (**C,D,J**–**L**). The scale bars represent 1 cm.

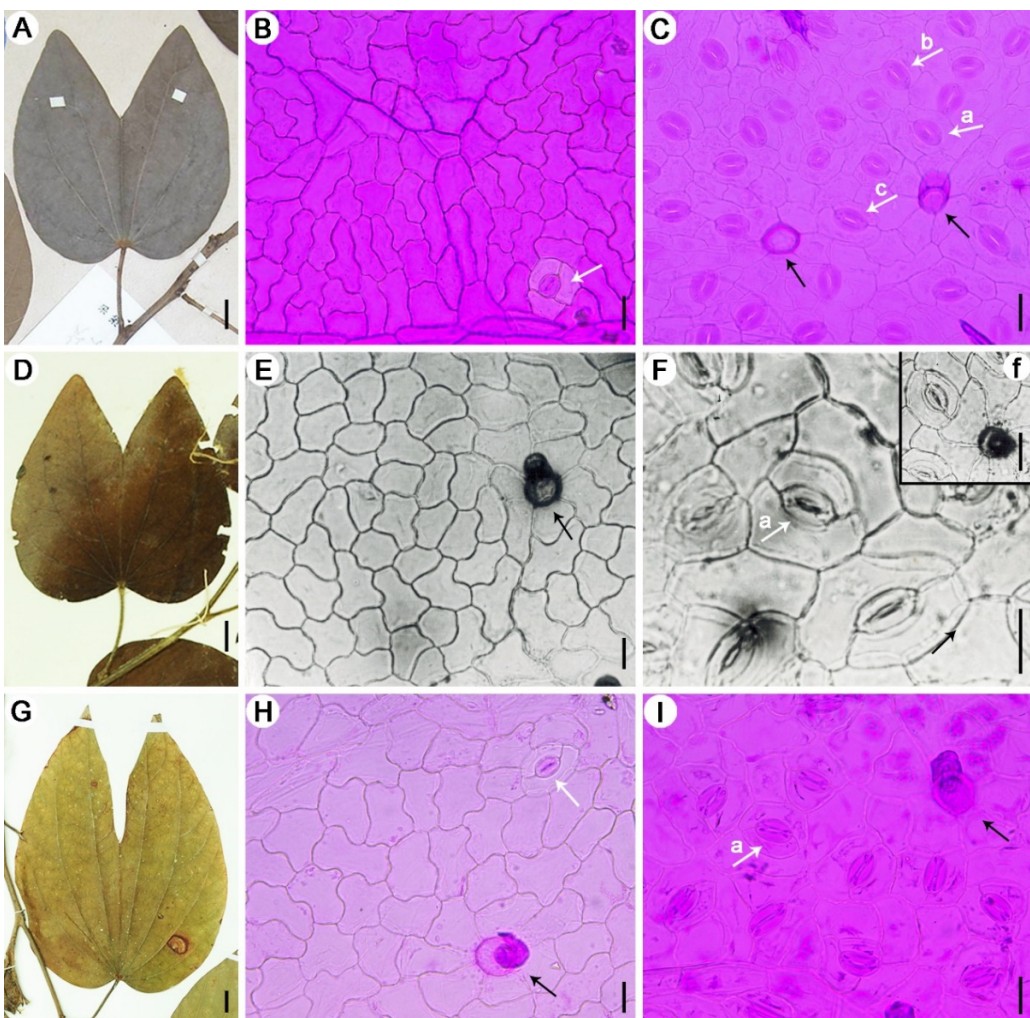

**Figure 8.** Leaf morphologies, adaxial (**B,E,H**) and abaxial (**C,F,I**) epidermises of *Bauhinia acuminata* L. (**A–C**) (IBSC, 0166379), *B. comosa* Craib (**D–F**) (HITBC, 0009647), and *B. esquirolii* Gagnep. (**G–I**) (HITBC, 0021422). The scale bars represent 1 cm for leaf specimens and 20 μm for epidermal images. The image f in (**F**) indicates a paracytic stomatal complex and a single-celled trichome base in abaxial epidermis. White arrows in (**B,H**) indicate few stomata in adaxial epidermis; white arrows in (**C,F,I**) indicate (a) paracytic stomatal complex, (b) tetracytic stomatal complex, and (c) sunken guard cells; black arrows indicate single-celled trichome bases.

**Table 1.** Leaf and fruit fossil records of *Bauhinia*.

| No. | Species | Type | Age | Locality | Reference |
|---|---|---|---|---|---|
| 1 | *Bauhinia wenshanensis* H.H. Meng et Z.K. Zhou | Leaf | Early Oligocene | Dashidong Town, Wenshan County, Yunnan Province, China | [11] |
| 2 | *Bauhinia larsenii* D.X. Zhang et Y.F. Chen | Leaf and fruit | Late Oligocene | Ningming County, Guangxi, China | [10,12] |
| 3 | *Bauhinia ningmingensis* Q. Wang | Leaf | Late Oligocene | Ningming, Guangxi, China | [10] |
| 4 | *Bauhinia cheniae* Q. Wang | Leaf | Late Oligocene | Ningming, Guangxi, China | [10] |
| 5 | *Bauhinia ningmingensis* Q. Wang | Leaf | Late Oligocene | Ningming, Guangxi, China | [10] |
| 6 | *Bauhcis moranii* Calvillo-Canadell et Cevallos-Ferriz | Leaf | Oligocene | Los Ahuehuetes, Tepexi de Rodríguez, Puebla, Mexico | [14] |
| 7 | *Bauhinia krishnanunnii* A. K. Mathur | Leaf | Early Miocene | Unmetalled way to Babu Mohalla, Dagshai Cantonment, Solan District, Himachal Pradesh, India | [16] |
| 8 | *Bauhinia fotana* F. M. B. Jacques | Leaf | Middle Miocene | Zhangpu, County, Zhangzhou City, Fujian Province, Southeast China | [15] |
| 9 | *Bauhinia ungulatoides* Y. X. Lin, W. O. Wong, G. L. Shi, S. Shen et Z. Y. Li | Leaf | Middle Miocene | Zhangpu, Fujian, China | [9] |
| 10 | *Bauhinia ecuadorensis* E.W. Berry | Leaf | Miocene | Loja Basin, Ecuador | [17] |
| 11 | *Bauhinia siwalika* R.N. Lakh. et N. Awasthi | Leaf | Middle Miocene-Pleistocene | Bhikhnathoree, West Champaran District, Bihar, India | [36] |
| 12 | *Bauhinia nepalensis* N. Awasthi et N. Prasad | Leaf | Middle Miocene-Pleistocene | Bhikhnathoree, West Champaran District, Bihar, India | [36] |
| 13 | *Bauhinia nepalensis* N. Awasthi et N. Prasad | Leaf | Middle Miocene-Pleistocene | Surai Khola beds, near SuraiKhola bridge, Surai Khola area, India | [37] |
| 14 | *Bauhinia siwalika* R.N. Lakh. et N. Awasthi | Leaf | Middle Miocene-Middle Pleistocene | Bhikhnathoree, West Champaran District, Bihar, India | [38] |
| 15 | *Bauhinia waylandii* R.W. Chaney | Leaf | Pliocene | Busano, Bugishu, District, Eastern, Province, Uganda | [18] |
| 16 | *Bauhinia potosiana* E.W. Berry | Leaf | Pliocene-Early Pleistocene | Potosi, Bolivia | [39] |
| 17 | *Bauhinia* sp. cf. *B. purpurea* L. | Leaf | Late Cenozoic | Mahuadanr, Palamau District, Bihar, India | [40] |

Note. Carpenter et al. [41] reported bilobed leaves from the Cenozoic of Australia and assigned these fossils to cf. Cercideae/Detarieae. It is clear that the veins diverge from the midvein and extend into the apex of the lobes in these fossils, which distinguishes them from *Bauhinia* in which two of the basal veins extend into the lobe apices. Moreover, Biagolini et al. [42] documented a fragment of a leaf, lacking apex and cuticle, from the Paleogene of Brazil. Their fossil seems to have palmate venation, a kind of venation pattern that exists in many families such as Malvaceae, Euphorbiaceae and Lauraceae. This makes the assignment of the leaf to *Bauhinia* superficial.

Morphotype 2 is similar to *B. purpurea* L., *B. viridescens* Desv., *B. tomentosa* L., and *B. racemosa* Lam. in terms of macroscopic morphology (Figure 9). However, the adaxial and abaxial epidermis of *B. purpurea* do not have trichome bases (Figure 9C,D), and so are different from those of morphotype 2 that possesses single-celled trichome bases (Figure 5G,K). The adaxial epidermal walls of *B. viridescens* are sinuolate (Figure 9G) whereas those of morphotype 2 are straight (Figure 5D–F). The abaxial epidermis of *B. tomentosa* has single-celled glandular trichome bases (Figure 9L), distinguishing it from morphotype 2 that has regular single-celled trichome bases (Figure 5G,K). The trichome bases for adaxial and abaxial epidermises of *B. racemose* are multicellular (Figure 9O,P) whereas those of morphotype 2 are unicellular (Figure 5G,K). In addition, *B. purpurea*, *B. viridescens*, and *B. tomentosa* have paracytic stomatal complexes (Figure 9D,H,L), and so are different from morphotype 2 with paracytic and tetracytic stomatal complexes (Figure 5H–J). Moreover, these four extant *Bauhinia* species exhibit a single-layered stomatal rim (Figure 9C,D,G,H,K,L,O,P), whereas morphotype 2 has a double-layered stomatal rim (Figure 5H–I). Overall, the four extant species above are different from morphotype 2 in their cuticular features. When compared to the fossil species, morphotype 2 is similar to *Bauhcis moranii* Calvillo-Canadell et Cevallos-Ferriz from the Oligocene of Mexico [14]. Due to no cuticular information in *Bauhcis moranii* and limited preservation of morphotype 2, we leave the nomenclature open for discussion.

Morphotypes 3 and 4, are similar to leaves and fruits of two species, i.e., *Bauhinia touranensis* Gagnep. and *B. damiaoshanensis* T. Chen (Figure 10). The two morphotypes possibly represent the same species, but this species is distinguished from those represented by morphotypes 1 and 2 because the fruit (morphotype 4) is different from those of the close recent relatives of morphotypes 1 and 2. When compared to fossils, morphotype 3 is different from any previously reported species. *Bauhinia larsenii* D.X. Zhang et Y.F. Chen from the late Oligocene of Ningming Basin, southern China [12], is the only fruit fossil assigned to the genus so far. However, morphotype 4 is banded with an acuminate stigmatic remnant, distinguishing it from *B. larsenii* which is elliptical with an acute stigmatic remnant. We here treat the morphotypes 3 and 4 as undetermined species.

To conclude, our fossils represent at least three species. Morphotype 1 is assigned to *B. wenshanensis*. Morphotype 2 constitutes the second species (*B.* sp.), and morphotypes 3 and 4 are possibly the third one (*B.* sp.).

### 4.2. The Diversification of Bauhinia in Southeastern Asia

Southeastern Asia is one of the diversity centers of *Bauhinia* (Figure 1). A recent molecular phylogenetic study suggests that the diversification of Asian *Bauhinia* can be traced back to the Paleocene (~60 Ma) [11]. However, Asian *Bauhinia* fossils are only known from the early Oligocene so far (Table 1). This forms a gap between the fossil evidence and molecular dating. Our finding is most likely the earliest reliable fossil records of *Bauhinia* in Asia, extending the existence of the genus to the late Eocene. *Bauhinia wenshanensis* has been reported from the early Oligocene of Wenshan, and four species, i.e., *B. larsenii* D.X. Zhang et Y.F. Chen, *B. ningmingensis* Q. Wang, *B. cheniae* Q. Wang, and *Bauhcis moranii* have also been found from the late Oligocene of Ningming Basin, Guangxi, China [9,11,12]. This provides evidence that *Bauhinia* apparently diversified in Asia in the Oligocene. In the Miocene and later periods, *B. krishnanunnii* A. K. Mathur comes from the early Miocene of India, *B. fotana* F. M. B. Jacques and *B. ungulatoides* Y. X. Lin, W. O. Wong, G. L. Shi, S. Shen et Z. Y. Li have been documented from the Middle Miocene Fujian province of China, and *B. siwalika* R.N. Lakh. et N. Awasthi and *B. nepalensis* N. Awasthi et N. Prasad from the middle Miocene to Pleistocene of India. This suggests further accumulation of species diversity occurred within the genus.

An interesting phenomenon for the extant distributional pattern of *Bauhinia* in southwestern China is that a small area can harbor many species. For example, 10 species have been found living in the Laojun Mountain area while 14 species have been recorded from Dawei Mountain, southeastern Yunnan [43,44], close to the locality that yielded fossils

in this study. Of primary interest is when this kind of pattern formed. The discovery of three *Bauhinia* species from the late Eocene Puyang Basin and four species in the late Oligocene Ningming Basin (Table 1) shows that the two basins once harbored multiple *Bauhinia* species. Therefore, the phenomenon of many *Bauhinia* species coexisting in a small area can now be traced back to at least the Paleogene.

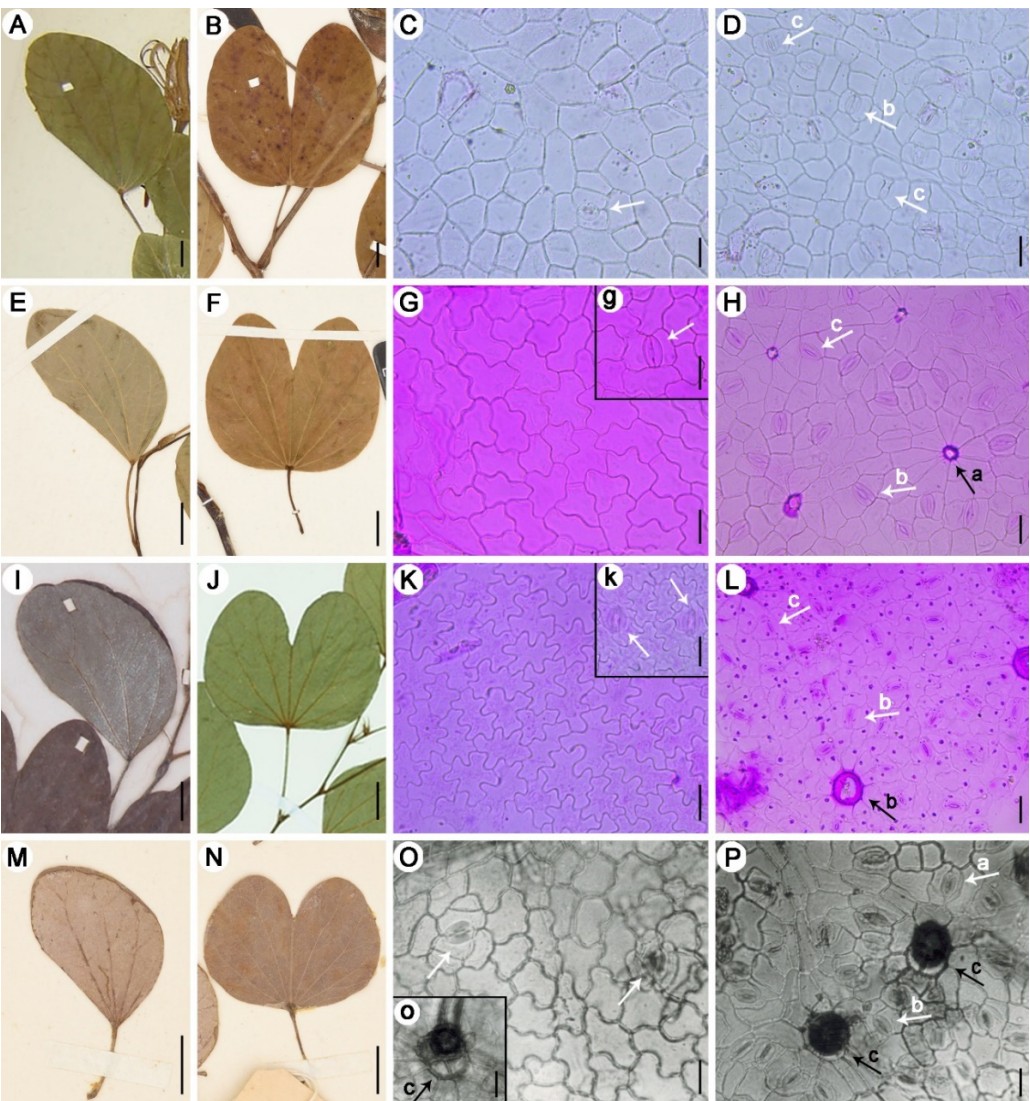

**Figure 9.** Leaf morphologies, together with adaxial (**C,G,K,O**) and abaxial (**D,H,L,P**) epidermises of *Bauhinia purpurea* L. (**A–D**) ((A) PE, 00327078; (B) KUN, 0125154), *B. viridescens* Desv. (**E–H**) (KUN 0169829), *B. tomentosa* L. (**I–L**) ((I) SYS, 00044882; (J) HITBC, 0021440), and *B. racemosa* Lam. (**M–P**) (KUN, 0125165). The scale bars represent 1 cm for leaf specimens and 20 µm for epidermal images. The image g in (**G**) indicates a stomatal complex of an adaxial epidermis. The image k in (**K**) indicates stomata of an adaxial epidermis. The image o in (**O**) indicates a trichome base of an adaxial epidermis. White arrows in (**C**), g, k, and (**O**) indicate stomata of adaxial epidermises; white arrows in (**D,H,L,P**) indicate (a) paracytic stomatal complex, (b) tetracytic stomatal complexes, and (c) sunken guard cells; black arrows indicate (a) a single-celled trichome base, (b) a single-celled glandular trichome base, and (c) multicellular trichome bases.

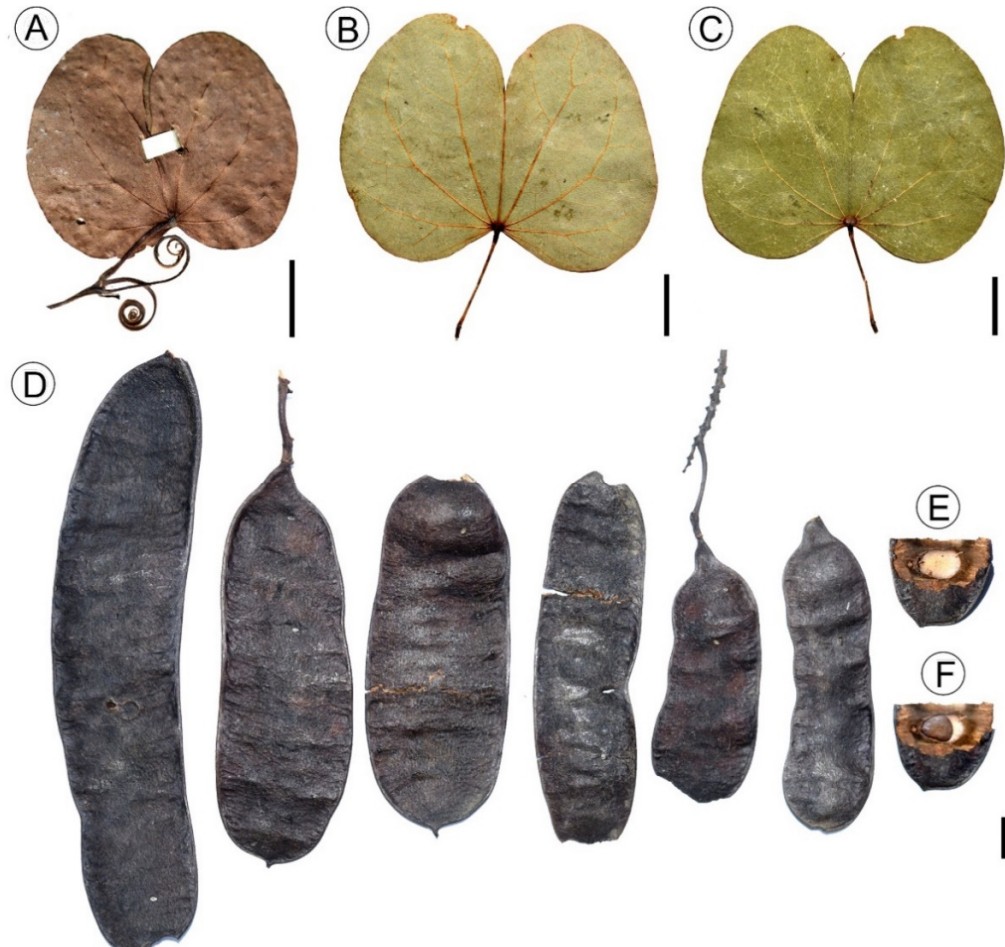

**Figure 10.** Morphology of *Bauhinia damiaoshanensis* T. Chen leaf (**A**) and *Bauhinia touranensis* Gagnep. leaves and fruits (**B**–**F**). The bars are 1 cm.

It is worth noting that although the Neotropics today host the largest diversity of *Bauhinia* species, few early fossil records have been documented there (Table 1 and note therein). Moreover, a recent molecular work points to a Neogene diversification of Neotropical *Bauhinia* species [11]. This scenario suggests that Asia is probably an ancient diversification center of *Bauhinia*, while the Neotropics is a more recent one, although this could result from under investigation of fossil records.

**Author Contributions:** Conceptualization, L.-B.J. and Z.-K.Z.; methodology, J.-C.Y. and J.-J.H.; software, J.L. and J.-C.Y.; validation, R.A.S., S.-T.Z. and Z.-K.Z.; formal analysis, J.-J.H. and L.-B.J.; investigation, J.-J.H. and Y.-J.H.; resources, P.Z.; data curation, P.Z.; writing—original draft preparation, L.-B.J. and J.-J.H.; writing—review and editing, R.A.S.; visualization, J.-J.H.; supervision, R.A.S., T.S. and Z.-K.Z.; project administration, L.-B.J.; funding acquisition, L.-B.J. and Z.-K.Z. All authors have read and agreed to the published version of the manuscript.

**Funding:** This work was supported by the National Natural Science Foundation of China (No. 31900194), the Science and technology Project of Yunnan Science and Technology Department (No. 202101AT070163, 202001AU070058), and the Chinese Academy of Sciences "Light of West China" Program and Yunnan Basic Research Projects (No.2019FB026).

**Institutional Review Board Statement:** Not applicable.

**Data Availability Statement:** All data are available in the text.

**Acknowledgments:** The authors are indebted to Tie-Yao Tu for identification of some extant *Bauhinia* species, Li Chen for collecting extant specimens of *Bauhinia* from southwestern Yunnan, China, and

Tao Deng, Xue-Dan Xie, Jing-Hua Wang, Ping Li, Xia Yang, Shi-Wei Guo, and Yu-Lan Yang for providing pictures of extant *Bauhinia* and help in morphological comparisons. We are also grateful to the Herbarium of Kunming Institute of Botany, Chinese Academy of Sciences (KUN) for providing access to the extant specimens.

**Conflicts of Interest:** The authors declare no conflict of interest.

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
