# Peer review of "Bauhinia (Leguminosae) Fossils from the Paleogene of Southwestern China and Its Species Accumulation in Asia"

_diversity, doi:10.3390/d14030173_

Round 1

Reviewer 1 Report

Dear authors, dear editor,

The authors present new Bauhenia leaf fossils from Yunnan.

This manuscript has the potential for publication in this journal. But there are several fundamental points of critique that need to be answered and taken care of before the manuscript can be considered to be accepted.

Major points of critique:

Age constraint:

For  age constraint the authors refer to a publication which is still under review. It is not possible for me, the reviewer, to check the correctness of the assumed age of the leaf fossils. Please include either the fossil range of the found Anthracotherium species as a supplement or include the age-constraining pollen information as a supplement. With the information given in the manuscript the reviewer can only check the stratigraphic range of Anthracotherium (Eocene to Miocene).

Morphotype diversity is not the same as species diversity!

I agree with the morphotypes you figure and distinguish. But I am not fully convinced that each morphotype represents a species of its own. The leaf variability that an extant species (and sometimes a single tree) shows may include all your morphotypes. I had a short look at the leaf diversity documented for Bauhinia acuminata. See the link below: https://www.gbif.org/occurrence/gallery?taxon_key=2953747

In this online collection of pictures several leaves would fit the three morphotypes. Please see for yourself.

A good example for leaf variability is the fossil species Quercus drymeja (Denk et al., 2017).

Quality of fossil cuticular LM micrographs:

The single cell trichomes were visible in the used LM micrographs. Concerning the stomata configuration, I could not distinguish the described features. The preservation maybe suboptimal. Please use in focus pictures or higher magnification of the described features. An alternative would be to show the original picture and next to it a picture with the stomata features enhanced or highlighted in colors or grey shades.

Concerning the use of "cf."

If you use Bauhinia cf. acuminata you indicate that this species is possibly a member of extant Bauhinia acuminata. The fossil is over 35 Myr old, that is a lot of time for evolution.

More fitting would be to name the specimens of this morphotype Bauhinia cf. wenshanensis. Especially, because you mention the strong similarities to this fossil species (line 230).

Language:

The manuscript should be checked and corrected by a native speaker with botanical or palaeobotanical background. Several passages could be formulated better.

Minor points of critique:

  • Please include coordinates for the locality.
  • Consistency: In the manuscript you give ranges in different ways:

Line 110: 28-34 mm long and 18-26 mm wide
Line 114: 1/4 – 1/3
Please change this to a consistent style in the entire manuscript.

  • Font size: I do not know if this was intentionally done, but the font size varies in the manuscript. This needs to be changed to be consistent.

I recommend major revision of the present manuscript. I will be available for additional review of the annotated manuscript.

References:

Denk, T., Velitzelos, D., Güner, H.T., Bouchal, J.M., Grimsson, F., Grimm, G., 2017b. Taxonomy and palaeoecology of two widespread western Eurasian Neogene sclerophyllous oak species: Quercus drymeja Unger and Q. mediterranea Unger. Rev. Palaeobot. Palynol. 241, 98–128.

Author Response

Response to reviewer #1

General comments 1 The authors present new Bauhinia leaf fossils from Yunnan. This manuscript has the potential for publication in this journal. But there are several fundamental points of critique that need to be answered and taken care of before the manuscript can be considered to be accepted.

Response: We thank reviewer #1 for the comments and suggestions. We have revised the manuscript accordingly. Please see our point-by-point responses below.

Comment 1 Age constraint: For age constraint the authors refer to a publication which is still under review. It is not possible for me, the reviewer, to check the correctness of the assumed age of the leaf fossils. Please include either the fossil range of the found Anthracotherium species as a supplement or include the age-constraining pollen information as a supplement. With the information given in the manuscript the reviewer can only check the stratigraphic range of Anthracotherium (Eocene to Miocene).

Response: Yes. We have added more information for the mammal fossil, Anthracotheriidae, which is comparable to the late Eocene Bothriogenys hui from Yunnan and B. orientalis from Thailand. Please see lines 86-97. We have received comments from the reviewers for the manuscript (Yang et al. under review). Both of the two reviewers suggested minor revision. See the comments from the reviewers in Figure_for_review.

Comment 2 Morphotype diversity is not the same as species diversity! I agree with the morphotypes you figure and distinguish. But I am not fully convinced that each morphotype represents a species of its own. The leaf variability that an extant species (and sometimes a single tree) shows may include all your morphotypes. I had a short look at the leaf diversity documented for Bauhinia acuminata. See the link below: https://www.gbif.org/occurrence/gallery?taxon_key=2953747

In this online collection of pictures several leaves would fit the three morphotypes. Please see for yourself. A good example for leaf variability is the fossil species Quercus drymeja (Denk et al., 2017).

Response: We agree with reviewer #1 that the variation of leaf morphology within one species or the overlap of leaf morphology between different species do exist and this is the reason why cuticle is important in the fossil identifications. The stomata of morphotype 1 have uni-layered stomatal rims whereas those of morphotype 2 have double-layered stomatal rims. Single or double layered stomatal rims are considered stable in intraspecific level (Botanical Journal of the Linnean Society 148: 39-56). Besides, the adaxial epidermal walls of morphotype 1 are sinuolate but these of morphotype 2 are straight although this could be intraspecific variation. Thus, morphotype 1 and 2 should represent different species. We failed to get cuticle from morphotype 3 after endeavoring for several times. However, when we put morphotype 3 and morphotype 4 (a fruit) together, they are very similar to those of Bauhinia damiaoshanensis T. Chen and B. touranensis Gagnep. which are distributed in the nearby regions where our fossils are collected. But morphotype 4 is apparently different from fruits of the closest relatives of morphotype 1 and morphotype 2. Therefore, we propose that the four morphotypes should represent at least three species. We have added several sentences to clarify this. See lines 264-277, 393-410. It is also worth noting that some pictures on the Gbif are misidentified and should be used with caution.

Comment 3 Quality of fossil cuticular LM micrographs: The single cell trichomes were visible in the used LM micrographs. Concerning the stomata configuration, I could not distinguish the described features. The preservation maybe suboptimal. Please use in focus pictures or higher magnification of the described features. An alternative would be to show the original picture and next to it a picture with the stomata features enhanced or highlighted in colors or grey shades.

Response: Yes, the preservation of our fossils may be suboptimal with indistinct guard cells, but the epidermal cells and stomatal apertures can be clearly seen. It’s also possible that the guard cells are too sunken to distinguish. In some extant species, for example, B. purpurea in Fig. 9D in our manuscript, the sunken guard cells are also not easy to distinguish. We have tried oil immersion objective in LM, SEM and fluorescence microscope to observe the fossil cuticle, but there are also no better micrographs with distinct guard cells. However, the patterns of stomata and subsidiary cells are not difficult to judge. Therefore, we have added line drawings to illustrate the stomatal complexes observed in the fossil cuticle.

Comment 4 Concerning the use of "cf." If you use Bauhinia cf. acuminata you indicate that this species is possibly a member of extant Bauhinia acuminata. The fossil is over 35 Myr old, that is a lot of time for evolution. More fitting would be to name the specimens of this morphotype Bauhinia cf. wenshanensis. Especially, because you mention the strong similarities to this fossil species (line 230).

Response: We have assigned morphotype 1 to Bauhinia wenshanensis as suggested. See line 135.

Comment 5 Language: The manuscript should be checked and corrected by a native speaker with botanical or palaeobotanical background. Several passages could be formulated better.

Response: Yes, the manuscript has been proofread by one of the coauthors, Bob Spicer, who is a native speaker.

Comment 6 Minor points of critique:

  • Please include coordinates for the locality.
  • Consistency: In the manuscript you give ranges in different ways:

Line 110: 28-34 mm long and 18-26 mm wide
Line 114: 1/4 – 1/3
Please change this to a consistent style in the entire manuscript.

  • Font size: I do not know if this was intentionally done, but the font size varies in the manuscript. This needs to be changed to be consistent.

Response: The coordinates of the fossil site have been added. The way that gives the ranges and the fonts are now in a way of consistency.

Reviewer 2 Report

Overall a well written account of some new Cecideae-like fossils from China. There are several points in ms presentation etc. noted on the annotated PDF which need addressing. The main issues I have are the absence of several key references, as well as the lack of detailed comparisons with other bilobed Cercideae (at least two of which are not mentioned), especially for their cuticles. At the very least there needs to be more justification as to why the fossils must be Bauhinia. There also needs to be discussion of the Paleogene Sth Amer Bauhinia fossil described in 2013 and what effect this may have on the overall conclusions about timing of evolution and dispersal etc. in the group.

Author Response

Response to reviewer #2

General comments 2 Overall a well written account of some new Cecideae-like fossils from China. There are several points in ms presentation etc. noted on the annotated PDF which need addressing. The main issues I have are the absence of several key references, as well as the lack of detailed comparisons with other bilobed Cercideae (at least two of which are not mentioned), especially for their cuticles. At the very least there needs to be more justification as to why the fossils must be Bauhinia. There also needs to be discussion of the Paleogene Sth Amer Bauhinia fossil described in 2013 and what effect this may have on the overall conclusions about timing of evolution and dispersal etc. in the group.

Response: Thank you for your positive comments. We have revised the manuscript accordingly. The mentioned references have been added or evaluated. For the comparisons with other bilobed Cercideae, that depends on how we adopt the classification scheme of Bauhinia L. In some recent treatments, Bauhinia L. is divided into eight genera including Bauhinia L. s.s., Barklya F. Muell., Gigasiphon Drake, Lysiphyllum (Benth.) de Wit, Phanera Lour., Piliostigma Hochst., Schnella Raddi, and Tylosema (Schweinf.) Torre et Hillc. Because the phylogenetic relationship within Bauhina + Griffonia + Brenierea clade has yet been fully resolved and the identification of Bauhinia leaf fossils possibly could not reach the Bauhinia L. s.s. level, we here adopt a traditional broad treatment of Bauhinia. See lines 51-62. Thus the mentioned Barklya and Lysiphyllum are actually included in the Bauhinia L. and we have not further compared our fossils with the two genera. As far as we know, there are no other bilobed leaves except that Bauhinia L. is morphologically identical with our fossils. We thank reviewer #2 to figure out Dilobia (Proteaceae) which also produces bilobed leaf. We have included this genus in the comparison. For the South American Paleogene Bauhinia, we have evaluated the B. aff. B. divaricate (see Brazilian Journal of Geology, 43(4): 639-652, Figure 4 A) preserved as a fragment without information of cuticle. We think the leaf fragment should have palmate venation, but this kind of leaves are found in many families such as Euphorbiaceae and Lauraceae. So we did not include the South American fossils in our manuscript.

Comment 7 Line 56 Biagolini, C. H., Bernardes-de-Oliveira, M. E. C., & Caramês, A. G. (2013). Itaquaquecetuba Formation, São Paulo basin, Brazil: new angiosperm components of Paleogene taphoflora. Brazilian Journal of Geology, 43(4), 639–652. "Bauhinia aff. B. divaricata"

Response: Yes, the South American "Bauhinia aff. B. divaricata" has been evaluated. It is a fragment of a leaf, lacking apex and cuticular information (see Figure 4 A in Brazilian Journal of Geology, 43(4), 639–652). The fossil seems to have palmate venation, but this kind of venation pattern could be found in many families such as Malvaceae, Euphorbiaceae and Lauraceae. So it may be not appropriate to be included in our study. Please also refer to our response to General comments 2.

Comment 8 Line 58 Carpenter, R. J., Goodwin, M. P., Hill, R. S., & Kanold, K. (2011). Silcrete plant fossils from Lightning Ridge, New South Wales: new evidence for climate change and monsoon elements in the Australian Cenozoic. Australian Journal of Botany, 59(5), 399–425. https://doi.org/10.1071/BT11037 Cenozoic (possibly latest Oligocene to mid–late Miocene) macrofossils from the Lightning Ridge include Bauhinia-like leaves from tribe Cercideae.

Response: Yes. The bilobed leaves in Carpenter et al. (2011) (Figs. 58-64) are well preserved and were assigned to cf. Cercideae/Detarieae (Fabaceae). In Bauhinia, two of the basal veins extend into the apex of the lobes (see BMC Evolutionary Biology (2015) 15:252 DOI 10.1186/s12862-015-0540-9). However, in those of the Carpenter et al. (2011), the veins diverged from the middle vein extend into the apex of the lobes. Therefore, we think the fossils in Carpenter et al. (2011) (Figs. 58-64) should not represent Bauhina. Please also refer to our response to General comments 2.

Comment 9 Line 180 Also Bauhinia-like Proteaceae. see Pole, M., & Bowman, D. M. J. S. (1996). Tertiary plant fossils from Australiaʼs ‘Top End’. Australian Systematic Botany, 9, 113–126.

Response: Important supplement. We have included Dilobia (Proteaceae) in the comparisons. Please see lines 236-237.

Comment 10 Line 187 Also the Australian genera Barklya and Lysiphyllum See discussion in Carpenter et al. (2011)

Response: As in this study we adopt a broad circumscription of Bauhinia, Barklya and Lysiphyllum are actually members of Bauhinia. Please also refer to our response to General comments 2.

Comment 11 Line 191 There is a problem here as you assume morphology alone is enough to place into a genus, whereas many leaf taxa can be extremely similar and need cuticular comparisons to support placement. This should definitely be the case for at least the other Cercideae genera with bilobed leaves

Response: The species with bilobed leaves in the Cercideae were placed in the traditional Bauhinia L. which was divided into eight small genera in some recent treatments. We agree with the reviewer that leaf morphology may be not enough to distinguish the eight genera. Therefore, we adopt a traditional broad circumscription of Bauhinia. This could be more appropriate. Please also refer to our response to General comments 2.

Comment 12 Line 194 Expand to include missing Aust bilobed genera

Response: This has been done by Lin et al. BMC Evolutionary Biology (2015) 15:252 DOI 10.1186/s12862-015-0540-9. We have cited this paper in the manuscript. Please also refer to our response to General comments 2.

Comment 13 Line 262 Taxon names in italics

Response: Yes, the whole manuscript has been checked.

Comment 14 Line 305 Rework in terms of the fossils in Biagolini et al. (2013)

Response: The reference Biagolini et al. (2013) has been evaluated. Please refer to our response to Comment 7.

Round 2

Reviewer 1 Report

Dear authors, dear editors,

The requests I had in my first review have been answered to full extent.

I have gone through the updated PDF. I have attached  an annotated PDF, in which have a few small comments (concerning punctation and phrasing).

I congratulate the authors to their findings and recommend the acceptance of the manuscript.

With kind regards

JM Bouchal, Darmstadt, 21st February 2022

Author Response

Response to reviewer #1

Dear authors, dear editors,

The requests I had in my first review have been answered to full extent.

I have gone through the updated PDF. I have attached an annotated PDF, in which have a few small comments (concerning punctation and phrasing).

I congratulate the authors to their findings and recommend the acceptance of the manuscript.

With kind regards

JM Bouchal, Darmstadt, 21st February 2022

Response: We thank reviewer #1 for positive comments. The suggested punctation and phrasing are corrected accordingly. Please see lines 22, 33, 34, 52, 71, 88, 92, 108, 109, 121, 128, 186, 190, 197, 206, 225, 229, 279, 280, 351 and 377.

Reviewer 2 Report

Overall, the updated ms is now improved, with only a few minor points noted on the annotated pdf. However, even though the authors may disagree with the identifications of the fossils reported from Australia and South America as Bauhinia-like, they CANNOT ignore them, especially given their comment about the Neotropics in the discussion (and the fact that any search for Bauhinia or Cercideae and fossil will likely find them). The remarks about these excluded fossils in their rebuttal need to be added to the discussion in the text about the overall fossil record and evolution of the group as as justification of why they are not being treated further here. 

Author Response

Comment 1 Overall, the updated ms is now improved, with only a few minor points noted on the annotated pdf. However, even though the authors may disagree with the identifications of the fossils reported from Australia and South America as Bauhinia-like, they CANNOT ignore them, especially given their comment about the Neotropics in the discussion (and the fact that any search for Bauhinia or Cercideae and fossil will likely find them). The remarks about these excluded fossils in their rebuttal need to be added to the discussion in the text about the overall fossil record and evolution of the group as as justification of why they are not being treated further here.

Response: We thank reviewer #2 for the suggestion. We agree that the evaluations of Carpenter et al. (2011) and Biagolini et al. (2013) are better to be added in the manuscript because this may be also a reference for the readers. We have added a note right behind the table to indicate why the Australian and South American fossils in Carpenter et al. (2011) and Biagolini et al. (2013) do not represent Bauhinia. Please see lines 284–290. We have also indicated the facts when mentioning the Neotropics. Please see line 377. The corresponding references were cited. Please see lines 401-403 and 409–411.

Comment 2 Line 93 suggests a late.

Response: Yes, “an” is now replaced by “a”.

Comment 3 Line 388 have not yet been reported for.

Response: Yes, “have yet been report in” is now replaced by “have not yet been reported for”.

Comment 4 Line 392 If you are going to exclude the Australian and South American reports then there has to be an explanation and justification of why in the text, not just in the rebuttal to reviewers.

Response: A note has been added for the table to comment the Australian and South American fossils. Please see lines 284–290 and our response to Comment 1.